# Interprofessional Educational Interventions to Improve Pharmacological Knowledge and Prescribing Competency in Medical Students and Trainees: A Scoping Review

**DOI:** 10.3390/pharmacy13050116

**Published:** 2025-08-27

**Authors:** Alec Lai, Viki Lui, Weiwei Shi, Brett Vaughan, Louisa Ng

**Affiliations:** 1Department of Medical Education, University of Melbourne, Parkville 3010, Australia; alecl@student.unimelb.edu.au (A.L.); louisan@unimelb.edu.au (L.N.); 2The Royal Melbourne Hospital, Parkville 3050, Australia; weiwei.shi@mh.org.au; 3School of Medical and Health Sciences, Edith Cowan University, Perth 6027, Australia; 4Faculty of Health, Southern Cross University, Lismore 2480, Australia

**Keywords:** interprofessional education, medical students, doctors, prescribing competency, pharmacology education, medication safety, scoping review

## Abstract

**Introduction**: Prescribing errors are the most common cause of preventable patient harm. In recent years, interprofessional education (IPE) has been increasingly utilised to improve knowledge and skills through promoting interprofessional collaboration. This scoping review aimed to evaluate the effectiveness of IPE interventions for pharmacological knowledge and prescribing skills in medical students and doctors-in-training. **Methods**: MEDLINE, EMBASE, CENTRAL, CINAHL, PsycINFO, ERIC and Scopus were searched on 18 February 2025 for studies published since 2020. Keywords included interprofessional education, medical student, medical trainee, pharmacology and prescribing. **Results**: Of the 2254 citations identified, 42 studies were included. There were four main types of IPE interventions: case-based learning, work-integrated-learning, didactic, and simulation and role-plays. Outcomes were spread across pharmacological knowledge, prescribing skills and interprofessional attitudes, and all studies reported one or more positive findings at Kirkpatrick IPE level 1, 2a, 2b, 3 or 4b. No study reported outcomes at Kirkpatrick IPE 4a. **Conclusions**: IPE interventions targeting pharmacology and prescribing are positively viewed by medical learners. IPE is effective in improving interprofessional attitudes and collaboration, as well as pharmacological knowledge and prescribing competency. Logistical challenges can be barriers to larger-group IPE implementation; nonetheless, IPE work-integrated learning in authentic clinical settings may overcome these challenges.

## 1. Introduction

Prescribing errors are the most frequent cause of preventable patient harm, with adverse medication events occurring in approximately 6.5 per 100 hospital admissions [1]. Twelve percent of adverse medication events are reported to be life-threatening, and one percent are fatal [1]. Medication-related errors are estimated to cause a financial burden of around USD 42 billion every year worldwide [2]. Of all prescribing errors made by doctors, doctors in their first year of practice are responsible for 91% of these errors [3], while the most experienced practitioners make the fewest [4]. Doctors-in-training are responsible for writing most of the medication prescriptions [5], particularly in hospital environments. However, the quantity of prescribing alone is not the sole contributor to prescribing errors. Many contributing factors have been identified by doctors, such as the work environment, knowledge gaps, or inadequate training [5]. Reducing medication errors is complex, and can be targeted with system-level approaches, judicious use of technology, and practitioner education. Currently, there are systems in place to help reduce medication errors such as electronic prescribing systems with built-in safety checks, and pharmacist-led medication reconciliation [6]. Pharmacist involvement has been found to reduce medication errors by 37% [7] suggesting a key role for these professionals in medication safety. In addition to these approaches to reducing medication errors, educational interventions should be a key component of any multi-faceted approach, as improving pharmacological knowledge and prescribing competence among medical students and doctors-in-training, has the potential to reduce the number of prescribing errors during the early practice years [3,8].

In recent years, interprofessional education (IPE) interventions have been increasingly utilised to improve pharmacological knowledge and prescribing competencies in both doctors [9] and medical students [10,11]. A recent scoping review by Shi et al. [8] investigated the effectiveness of educational interventions aimed at enhancing pharmacological knowledge and prescribing competency among medical students and reported generally positive outcomes at Kirkpatrick level 1 (learner reactions such as satisfaction) and level 2 (learning, such as knowledge gains). Interprofessional education was one of the approaches identified in the review and positive outcomes at Kirkpatrick level 1 were found [8]. These findings align with interprofessional collaboration becoming an increasingly recognised cornerstone of efficient and high-quality healthcare systems [12]. In educational settings, IPE brings learners from at least two professions to “learn about, from and with each other” to improve health outcomes [12]. In a systematic review relating broadly to IPE interventions, Guraya et al. [13] found that IPE interventions improved learners’ knowledge, skills and attitudes towards collaborative practice, in addition to enhancing job satisfaction, promoting effective problem-solving, and reducing professional stereotypes. In the context of medication prescribing, IPE interventions can focus on strengthening pharmacological knowledge and prescribing competency, and/or skills that support effective interprofessional collaboration [14].

In health professions education, much of IPE has been focused on medical students with increasing interest in other health professions. There is now an increasing interest in IPE outcomes for doctors-in-training. In 2025, the Australian Medical Council, the national standards body for medical education and assessment of medical professionals in Australia, introduced a National Framework for Prevocational Medical Training in 2025 which incorporated Entrustable Professional Activities (EPAs), assessing learning through IPE-informed principles [15]. Whilst EPAs are typically classed as assessments, they can also support learning by guiding feedback and reflection during training. Hence, involving interprofessional colleagues in EPA assessments align with IPE approaches. Emerging literature appears to support IPE and to suggest that IPE for doctors-in-training may have a direct impact on patient care. For example, a recent intervention involving peer-to-peer discussions between family medicine residents and pharmacy students resulted in a marked reduction in opioid prescribing following the programme [16]. Despite these promising outcomes, there remains some uncertainty around the acceptance of IPE for doctors-in-training. A scoping review examining IPE-based EPAs found that medical and nursing learners were uncertain about these EPAs and preferred receiving feedback from peers in their own discipline over receiving feedback from members of a different profession [17]. This preference for feedback from a profession congruent supervisor in doctors has also been described in another review [18]. This highlights the importance of incorporating training that encourages learners to recognise and appreciate the value of feedback from other professions, which would foster more effective interprofessional learning and collaboration.

Further, most IPE interventions in pharmacology and prescribing to date appear to have reported only Kirkpatrick level 1 and 2 outcomes—Shi et al. [8] reported no outcomes at Kirkpatrick Level 3 (behavioural change) or 4 (impact on organisational practice or patient outcomes). This may have been due to the focus on medical students in the review, who typically do not prescribe [8]. Nevertheless, if the ultimate objective of such education is to improve patient outcomes, then a significant gap in the literature remains. Similarly, a 2023 systematic review investigating the effectiveness of IPE in reducing medication safety found that few studies measured medication safety skills or outcomes [19]. Together, these knowledge syntheses suggest there are opportunities to better understand the outcomes associated with educational interventions, including IPE, in impacting prescribing practice and patient outcomes.

Given the increasing interest and implementation of IPE interventions aimed at improving pharmacological knowledge, prescribing competencies, and interprofessional collaboration in healthcare, an evaluation of contemporary literature can provide new insights. These insights can support medical educators in designing future IPE interventions. Therefore, this scoping review aims to comprehensively describe the range of IPE interventions involving learners in the medical profession (medical students and doctors-in-training) and their outcomes at all Kirkpatrick levels, including outcomes, where available, on patient care.

## 2. Materials and Methods

This scoping review was conducted in accordance with the principles outlined in the Joanna Briggs Institute methodology for scoping reviews [20] and reported in accordance with PRISMA (Preferred Reporting Items for Systematic reviews and Meta-Analyses) [21]. This scoping review protocol was registered 31 August 2024, on Open Science Framework registries and can be accessed here https://osf.io/fw8qb (accessed on 31 August 2024). The reviewing process was facilitated using Covidence [22], an online systematic review management tool. ChatGPT-4 (OpenAI, San Francisco, CA, USA) was utilised to improve the language in this review.

### 2.1. Search Strategy

The search strategy aimed to identify published and unpublished primary sources of evidence. An initial search of Ovid MEDLINE was conducted to identify relevant articles on this topic and to inform the final search strategy (AL with librarian assistance). A database search of MEDLINE, EMBASE, CENTRAL, CINAHL, PsycINFO, ERIC and Scopus was conducted on 18 February 2025. Keywords and Medical Subjective Headings (MeSH) or equivalent used included interprofessional education, interprofessional relations, competency-based education, medical student, medical school, medical trainee, resident, pharmacology and prescriptions. The search strategy created for MEDLINE was adapted for the other databases (see Appendix A). All databases were searched from January 2020 to 18 February 2025, as a previous similar scoping review had included studies published up to 2020 [8]. Therefore, to avoid duplication and to ensure contemporary literature was included, the search was narrowed to commence from 2020. Reference lists from included studies were hand-searched for additional studies. Grey literature was searched using OpenGrey (http://www.opengrey.eu accessed on 18 February 2025).

### 2.2. Inclusion and Exclusion Criteria

A PICO framework was developed to inform the inclusion and exclusion criteria. Primary research studies written in English were included if:

Population: Participants were learners in the medical profession, defined for the purposes of this review as medical learners, which comprised medical students (individuals undertaking a university-level degree in medicine) or doctors-in-training (junior doctors, such as interns or residents who have not yet completed specialist training).

Intervention: The educational content primarily focused on pharmacological knowledge or medication prescribing skills. Any aspect of prescribing could be targeted, including medication selection, safety, or communication skills related to the prescribing process. In authentic clinical settings, interventions were sometimes complex, interdisciplinary and tailored to patients. In such cases, it was not always possible to clearly delineate components of the intervention. Therefore, educational content on pharmacology was assumed, if required, based on reported outcomes. For example, if the intervention involved developing a discharge summary through collaborative efforts between medical and pharmacy professionals and the outcomes included medication-related outcomes, then the intervention was considered to have had a focus on pharmacological knowledge or medication prescribing skills even in the absence of an explicit description. In addition to the intervention focus, the intervention needed to be interprofessional in nature, defined as an educational activity involving interaction between at least two different healthcare professionals, either among learners or between learners and educators [23]. For example, there could be a combination of medical and pharmacy students learning together under the instruction of a healthcare professional, or medical professionals receiving education from a different health professional such as a pharmacist.

Comparison: No education or single-professional educational interventions.

Outcomes: Outcomes classifiable under Kirkpatrick’s expanded IPE learner outcomes (see Table 1) [24].

Exclusion criteria were as follows:Conference papers, abstracts, opinion letters, and commentaries.Studies where outcomes were not clearly attributable to medical learners.Studies involving osteopathic medical students were excluded, as the role and training of osteopaths in the United States differ significantly from other countries, where osteopathy is typically regarded as an allied health profession [25].

Titles and abstracts were independently screened by two reviewers (VL and AL) according to the pre-determined selection criteria. Full-text articles were retrieved and screened for inclusion. Discrepancies were discussed with a third reviewer (LN), and the final decision was made by consensus.

### 2.3. Data Extraction

Data was extracted from studies included in the scoping review by one reviewer (AL) using a data extraction tool developed by the authors. This tool was piloted on a sample of five studies and refined iteratively to ensure clarity and comprehensiveness. The following data were extracted from the studies included (see Appendix A):

Study design, country of study and year of publication;

Participants—Numbers, profession of the participants (for learners and for educators) and demographic background;

Intervention—Description of intervention and control (if relevant);

Outcomes—Measures, timepoints and results;

Any uncertainties that arose were resolved through discussion with two other reviewers (LN and VL). Where required, authors of papers were contacted to request missing or additional data.

## 3. Results

A total of 2254 studies were identified through the initial search. Following screening, 249 full-text studies were retrieved and further assessed for eligibility. Of these, 42 studies met the inclusion criteria (see Figure 1). Eighteen of the 42 studies (43%) were conducted in the United States of America.

There was a mix of study designs, including 22 pre- and post-intervention studies [14,26,27,28,29,30,31,32,33,34,35,36,37,38,39,40,41,42,43,44,45,46], 13 post-intervention studies [47,48,49,50,51,52,53,54,55,56,57,58,59], four pre- and post-intervention studies with a control group [60,61,62,63], two retrospective pre- and post-intervention studies [11,16], and one randomised controlled trial [64]. In total, there were 3282 learners (excluding three studies that did not specify participant numbers), comprising 2932 medical students and 350 doctors-in-training. Twenty-seven studies included medical students, 15 included doctors-in-training, and one study included both. Studies included between three and 386 medical participants. Other healthcare professionals involved as learner participants included a range of fields such as pharmacy, nursing, social work, sports science, addiction science and physician assistant. Outcomes of studies focused primarily on interprofessional collaboration, pharmacological knowledge or prescribing skills. Most studies reported only immediate post-intervention outcomes, but eight studies included delayed outcomes ranging from four days to two years.

Most studies featured interventions where interprofessional collaboration occurred between learners. Alternatively, 13 studies included interventions where either pharmacists or pharmacy students instructed medical learners [31,32,33,34,35,36,37,38,46,57,59,60,64]. Pharmacists were often in a direct teaching role, conducting workshops, facilitating case discussions or providing feedback on prescribing behaviour. These 13 studies reported Kirkpatrick IPE level outcomes at all levels, including at the higher levels. At Kirkpatrick IPE levels 1 and 2, seven studies reported learner satisfaction and improvements in confidence [31,32,34,35,36,37,59], and five studies reported significant improvements in pharmacological knowledge [31,34,37,59,60]. As for Kirkpatrick IPE levels 3 and 4 outcomes, two studies reported changes to prescribing behaviour [35,38], and four studies reported direct improvements to patient care post intervention [33,36,46,64]. Most studies that included pharmacist-led interventions focused on pharmacological knowledge and prescribing rather than on improving interprofessional collaboration outcomes per se [33,34,35,36,37,38,46,59,64]. Only four studies assessed changes in perceptions of medical students towards pharmacists [31,32,57,60], and of these, only one study found a significant improvement [60]. Of the studies that did not report significant changes, medical learners either already had a high baseline positive perception of pharmacists [31,32] or no statistical analysis was performed [57].

### 3.1. Assessment Outcomes

Assessment outcomes were classified into Kirkpatrick’s expanded IPE learner outcomes [24]. Kirkpatrick IPE outcomes were reported across all levels (1, 2a, 2b, 3 and 4b). Of the 42 included studies, 31 studies (73.8%) included a subjective student evaluation of the intervention (Kirkpatrick IPE 1) using feedback surveys, 5-point Likert scale questionnaires or focus group interviews. Of these, 21 studies measured student satisfaction with the intervention, nine assessed students’ confidence after the intervention and six reported students’ views of interprofessional collaboration. Thirty of the 42 included studies (71.4%) reported Kirkpatrick IPE 2 outcomes. Twenty-two (52.4%) of these studies reported level 2a outcomes by assessing attitudes towards IPC with surveys or interviews, while 11 studies (26.2%) utilised objective measurements, including knowledge tests such as multiple-choice questions (MCQ), objective structured clinical examinations (OSCEs) and measurements of prescribing skills by assessing the quality of discharge summaries and completion of prescribing cases. Of all the studies included in this review, two studies assessed the effects of IPE intervention on changing participant behaviour (Kirkpatrick IPE 3) and eight studies reported level 4b outcomes with direct benefits to patients. All included studies demonstrated a positive result for at least one assessed outcome. Twenty-two studies (51.2%) assessed for statistically significant differences. No studies reported level 4a outcomes.

### 3.2. Types of Interventions

Simulation and role-plays (ten studies, 23.8%), and work-integrated learning (ten studies, 23.8%), were the most reported interventions. Seven studies (16.7%) utilised case-based learning and five studies (11.9%) employed didactic learning. Ten studies (23.8%) used a combination of interventions, where a single dominant approach was not utilised. Typically, combination interventions included at least two or three of the following intervention types: case-based, simulations and role plays, work-integrated and didactic learning. While no e-learning interventions were described in the included studies, eight studies (19%) used an online medium to deliver case-based [53,56], didactic [28,30], simulation and role-play [29,52] or multicomponent learning interventions [54,55].

### 3.3. Simulation and Role-Plays

Simulation and role-plays were one of the most utilised interventions to deliver interprofessional education, with 10 studies (23.8%) investigating its effectiveness [14,29,32,37,39,40,41,52,61,62]. Nine of the 10 studies utilising simulations and role plays involved medical students [14,29,32,37,39,40,52,61,62], possibly because medical students are generally not allowed to prescribe.

Simulations were delivered through a variety of formats, including face-to-face workshops [14,32,37,39,40,61], hands-on use of medical devices [32,41] and simulations delivered online [29,52,62]. Learning activities featured standardised patient encounters, structured roleplays between interprofessional learners and a cloud-based simulation, where pairs of pharmacy and medical students managed a virtual patient case over a real-time simulation software. Eight of the 10 studies reported subjective (Kirkpatrick IPE 1) outcomes such as satisfaction and confidence and all eight found consistent improvement immediately post intervention [14,29,32,37,41,52,61,62]. Objective assessments such as knowledge tests, were conducted by four studies and all reported statistically significant improvements in pharmacological knowledge [14,37,40,41]. Five studies reported changes in interprofessional perceptions of medical learners. Of these, four found significant improvements in interprofessional attitude surveys post intervention [29,39,61,62] but not the study by Carroll and Hanrahan [32]. The latter reported no significant changes in medical students’ attitudes towards interprofessional administration of vaccines [32]. Importantly, it should be noted that this intervention primarily focused on practical vaccination skills and procedural training, which could have been less likely to shift interprofessional attitudes. Kirkpatrick IPE level 4b outcomes were reported by Gannon et al. [39] where a five-hour interprofessional simulation with debrief and “huddle” components resulted in findings that longer huddle times were associated with less patient harm.

In terms of barriers to the optimal implementation of IPE simulation interventions, logistical challenges were identified in several studies with varying impact. Heier et al. [62] reported an uneven ratio of medical to nursing students, with too few nursing students. Although this resulted in nursing students having to take on multiple roles, no negative consequences were reported [62]. Learners in the study by Jung et al. [61] who participated in a four-hour IPE intervention found the four-hour duration insufficient. Additional or longer sessions were not feasible primarily due to discrepancies in academic timetables which caused challenges in implementing multiple sessions [61].

Typically, simulations were synchronous in delivery, but one study implemented a real-time cloud-based simulation to facilitate asynchronous interprofessional simulation [52]. Learners reported this programme to be easy to use; however, medical students rated interactions with pharmacy students as moderate (equivalent to 3/5 on 5 point scale) using this programme [52], in contrast to the synchronous simulation interventions where interactions were typically highly rated [37,61]. Two other studies conducted simulations over Zoom [14,29]. The use of this video conferencing platform appeared acceptable in terms of the quality of interactions as these studies reported positive outcomes in student confidence [14,29], increased understanding of pharmacists’ role [29] and improved knowledge [14].

### 3.4. Work-Integrated Learning

Ten studies (23.8%) utilised work-integrated learning [16,26,27,36,46,49,50,51,57,60], defined as educational interventions conducted in authentic clinical or community-based care environments involving real patients to improve prescribing skills and interprofessional collaboration. Settings ranged from inpatient care to outpatient visits, telehealth consultations and student-led clinics. Notably, all work-integrated learning activities focused on shared care planning, collaborative decision making or feedback-driven learning. Five of the ten studies that assessed satisfaction reported positive outcomes [16,36,49,51]. Across seven studies [26,27,49,50,51,57,60], work-integrated learning was consistently found to improve interprofessional attitudes in medical learners, with three studies reporting statistically significant improvements [26,27,60]. Out of the ten studies that featured work-integrated learning, only one study included a control group to compare knowledge outcomes (Kirkpatrick IPE 2b) [60]. Students in the intervention group demonstrated a significant improvement in pharmacotherapy knowledge, while there were no significant changes in the control group [60]. Notably, four studies reported Kirkpatrick IPE 4b outcomes [16,36,46,50]. Both Reumerman et al. [50] and Aquino et al. [16] implemented student-led patient medication reviews and students were able to implement 68 medication changes and reduce average morphine dosage, respectively. Similarly, in the study by Cooper and Fitzpatrick [46], pharmacist-written emails containing prescribing tips and advice, informed by prescribing mistakes on the ward, were sent to doctors-in-training with a significant reduction in prescribing errors observed across both medical and surgical wards. Chang et al. [36] conducted a study in which an interprofessional team provided feedback and educated residents on hyperglycaemia management, resulting in a 25% reduction in patients’ blood glucose levels.

### 3.5. Case-Based Learning

Seven studies featured case-based learning (CBL) [14,31,47,48,53,56,59]. Student satisfaction was high across six studies [14,31,48,53,56,59], and improvements in attitudes and perceptions towards interprofessional collaboration were consistently observed among five studies evaluating CBL [14,31,47,48]. Despite this, Wilson et al. [14] found that, compared with other health professionals (pharmacy, nursing, addiction studies and social work students), medical students self-reported significantly lower improvements in pharmacological knowledge, and were least likely to find IPE beneficial for their education. Wilson et al. reported that this finding was consistent with a previous study which reported that students with a biomedical science background were less likely to be satisfied with IPE compared to those with a psychosocial background [65]. Two studies reported significant improvements in knowledge following the intervention (Kirkpatrick IPE 2b), and these were the highest Kirkpatrick level outcomes reported by studies utilising CBL [31,59]. One of the two studies assessed knowledge outcomes beyond the immediate post-intervention phase, and reported significant improvement compared to the national average when assessed in the year following post intervention [59].

### 3.6. Didactic Learning

Five studies examined the effectiveness of didactic learning [28,30,34,38,64] and reported this to be valuable for learning, with high learner satisfaction and perceived usefulness [28,30,34]. Two studies reported significant improvements in pharmacological knowledge immediately after the intervention [30,34], and Ring et al. showed a significant reduction in prescribing errors up to 3 months after intervention [38]. However, Ring et al. also measured the error rate at 4 months post intervention and found that prescribing errors had returned to pre-intervention baseline levels. The only randomised controlled trial included in this review featured a didactic intervention, and reported significant reduction in opioid prescriptions in the intervention group (15% reduction), with an adjusted incident rate ratio of 0.52 [64]. The control group also had a significant decrease in opioid prescriptions (6.4% reduction), likely attributable to the Hawthorne effect, where the exchange of knowledge between study groups, or institutional-wide policy changes improves performance across both intervention and control groups [64].

### 3.7. Multicomponent

Ten studies employed a combination of the aforementioned interventions, without a single more dominant approach [11,33,35,43,44,45,54,55,58,63]. These interventions were rated positively by students [45,54,55,63] and reported improved confidence [35,43,55]. Work-integrated learning was most frequently combined with at least one other intervention, with seven studies incorporating it into their design [11,33,35,43,44,45,63]. Similarly to the studies where work-integrated learning was the dominant approach, these studies were more likely to measure higher-level Kirkpatrick IPE outcomes. Two of these seven studies included a didactic learning component prior to receiving feedback on prescribing in clinical settings [33,63]. Both studies reported positive Kirkpatrick IPE 4b outcomes with statistically significant reductions in patient medication errors [33,63]. Naccarato et al. combined work-integrated learning with didactic and case-based learning and found doctors-in-training took more thorough medication histories (Kirkpatrick IPE 3) post intervention [35]. Most multicomponent studies measured changes in interprofessional attitudes and reported positive improvements [11,43,44,45,54,58].

## 4. Discussion

To the best of our knowledge, this is the first scoping review to evaluate the effectiveness of IPE interventions on pharmacological knowledge and prescribing competency in medical students and doctors-in-training. The 42 included studies described a range of educational interventions, including simulation and role-plays, case-based learning, work-integrated learning and didactic learning. All Kirkpatrick IPE levels, apart from level 4a, were reported and outcomes were largely positive across studies. IPE interventions featured interprofessional collaboration, most commonly between medical, pharmacy and nursing students. Across all studies, seven different healthcare professionals were involved, either as learners or facilitators.

The results of this review are consistent with previous studies. Similarly to previous reviews on IPE for health professions students, almost all included studies reported positive Kirkpatrick level 1 and 2 outcomes [8,66,67]. In contrast, where there was previously an absence of Kirkpatrick IPE level 3 and 4 outcomes in the literature [8,66,67], this review reported positive outcomes at both Kirkpatrick IPE level 3 and 4b. This is likely due to this scoping review’s broader scope of learners at different levels of training with the inclusion of doctors-in-training. Of the ten studies that reported level 3 or 4b outcomes, five of these were pharmacist-led interventions and learners were all doctors-in-training [33,35,36,38,64]. Compared to medical students, doctors-in-training are responsible for prescribing medications, making it more feasible to assess changes in prescribing skills and patient outcomes in real life clinical settings. This is reflected in a recent systematic review that found pharmacist-led interventions in the workplace resulted in a reduction in medication errors, with an odds ratio of 0.38 [68]. Although medical students are typically not allowed to prescribe, it may still be worth considering the types of educational interventions that could be applied in a safe but authentic manner to improve learning. For example, in the medication review programme for geriatric patients conducted by medical students, Reumerman et al. [50] reported that this was both educational and resulted in improved patient management.

Across the range of interventions, it appears that work-integrated learning lent itself most easily to assessing patient outcomes, likely as interventions were often already patient-focused. Of the eight studies that reported Kirkpatrick 4b outcomes, seven utilised work-integrated-learning principles in their IPE interventions [16,33,36,46,50,63,64]. Learners were involved in real patient care contexts where patient management decisions and interprofessional collaboration were realistic and consequential. Work-integrated learning IPE interventions were also likely to enhance student agency and self-regulation [69] and support career development by building personal attributes [70]. The two studies that reported Kirkpatrick IPE 3 outcomes implemented IPE interventions in a clinical context that aimed to modify prescribing behaviours [35,38]. While the authentic context made it more relevant to assess changes in prescribing skills, the positive outcomes observed likely also reflected the benefits of situated learning theory, which suggests that learning is more effective when it occurs in the environment where the skills being taught are applied [71]. Therefore, IPE conducted in an authentic clinical setting likely supports the translation of learning into outcomes that ultimately benefit patients.

Although positive outcomes were widely reported with IPE approaches, several challenges were highlighted. Logistical challenges were common and included difficulties coordinating timetables between disciplines [61] and uneven ratios between interprofessional groups [62]. These logistical challenges, particularly the coordination of schedules, have also been highlighted in a scoping review published in 2024, which evaluated barriers to implementing IPE for healthcare students [72]. Logistical issues may risk undermining the perceived value of IPE among students, especially when sessions are perceived to be disorganised or affected by low interprofessional participant attendance. The successful implementation of IPE requires efficient resource management and addressing logistical challenges through collaboration among staff members [72]. Notably, these types of barriers were not reported by the included studies which described work-integrated learning approaches. Incorporating IPE opportunities into existing collaborations within clinical settings likely obviates such barriers although the clinical setting itself would have its own challenges, including the provision of adequate supervision. In addition to logistical challenges, the current review also identified that some medical learners were less satisfied with IPE interventions when compared to other learners. In two studies where attitudes towards IPC were assessed and compared between different health learner groups, medical learners scored lowest [14,52]. These findings may reflect the influence of professional hierarchies within certain healthcare systems, where doctors may be perceived to occupy a higher position in the hierarchy, as described by Syahrizal et al. [73] and Omura et al. [74] in Indonesian and Japanese healthcare systems, respectively.

In terms of the use of online learning, it was interesting to note that there were no studies describing e-learning found. The development of e-learning in pharmacology and prescribing could present a good opportunity for interprofessional input, though it would require significant effort for e-learning to be interactive in nature. As for use of an online medium to facilitate educational interventions, there were a couple of issues noted. Firstly, medical students were less satisfied when this was used to allow interactions with pharmacy students, finding that it was harder to build interprofessional rapport compared to face-to-face sessions [52]. Secondly, spontaneous interactions were more challenging, thus requiring more effective facilitation to ensure better engagement with IPE activities and adequate support in collaborating with other professions [56]. These would be useful considerations given the increasing use of technology.

Limitations of this review included the challenges of developing a search strategy which allowed IPE to be identified across a range of medical learners (both medical students and doctors-in-training). The search strategy was refined repeatedly and iteratively until such studies were identified. Nonetheless, it is possible that the search strategy used may still not have captured all available studies. There was a low threshold for full text reviews to be conducted in order to optimise the identification of relevant studies. Further, the included studies had significant methodological weaknesses such as lack of randomisation, blinding and use control groups. Heterogeneity of study designs and outcomes also did not allow for comparisons across studies. Finally, all studies reported at least one positive outcome, suggesting a possibility of publication bias, although positive outcomes are common in educational interventions due to the nature of these interventions and a grey literature search was conducted to mitigate this.

In terms of implications for educators, this review has demonstrated that IPE is both readily received by medical learners and effective at improving pharmacological knowledge, prescribing competency and interprofessional collaboration, including at level 3 and 4 Kirkpatrick IPE outcomes. Educators should strongly consider work-integrated learning approaches delivered in authentic clinical environments where possible given their stronger association with direct patient outcomes. Such approaches may also reduce logistical barriers associated with large groups of interprofessional learners. For interventions conducted outside of clinical settings, forward planning is essential to address organisational and scheduling difficulties and to ensure the effective delivery of IPE. Further, education that is largely knowledge based rather than competency based may not be sustained [38] and may therefore benefit from additional refresher interventions to prevent knowledge and skill degradation. Additionally, prescribing errors are typically multifactorial. Ring et al. reported students continued to make medication errors related to decision making despite undertaking a prescribing skills focused intervention [38]. A multi-faceted approach, such as having a best practice advisory to remind prescribers which complements an education approach, should therefore be considered.

## 5. Conclusions

In conclusion, this scoping review demonstrates that interprofessional education is an effective strategy for improving medical learners’ pharmacological knowledge, prescribing competency, and interprofessional collaboration. Work-integrated learning approaches were associated with the highest Kirkpatrick IPE outcomes, while logistical challenges emerged as the most significant barrier to effective implementation of IPE.

## 6. Future Directions

This scoping review has highlighted gaps in the current literature, which should be addressed in future research. There is a need for robust methodologies with control groups to identify the benefits and differences between IPE and single-profession education. Despite higher Kirkpatrick IPE outcomes having been reported, there remains an absence of studies reporting Kirkpatrick IPE 4a outcomes, as current studies are focused on making changes at the individual level. While some studies have reported delayed outcomes, more longitudinal outcomes are required to determine if knowledge and skills obtained from IPE are sustained, or, in the case of medical students, whether there are positive outcomes later when they become prescribers. Finally, the use of standardised assessment outcomes would allow for better comparisons. For example, the Student Perceptions of Interprofessional Clinical Education–Revised (SPICE-R or SPICE-R2) could be used [75]. Such instruments have been shown to be reliable and to have construct validity [75]. They capture interprofessional teamwork and team-based practice; roles and responsibilities for collaborative practice; and patient outcomes from collaborative practice and have been developed specifically for medical and pharmacy students, and in the case of SPICE-R3, to include other healthcare professionals, such as nursing students [76]. As for the selection of measurement tools for Kirkpatrick IPE 4b outcomes, a framework which underpins the principles of safe, person-centred and quality use of medicines such as the one outlined by the National Prescribing Service could be used to inform this [77].

## Figures and Tables

**Figure 1 pharmacy-13-00116-f001:**
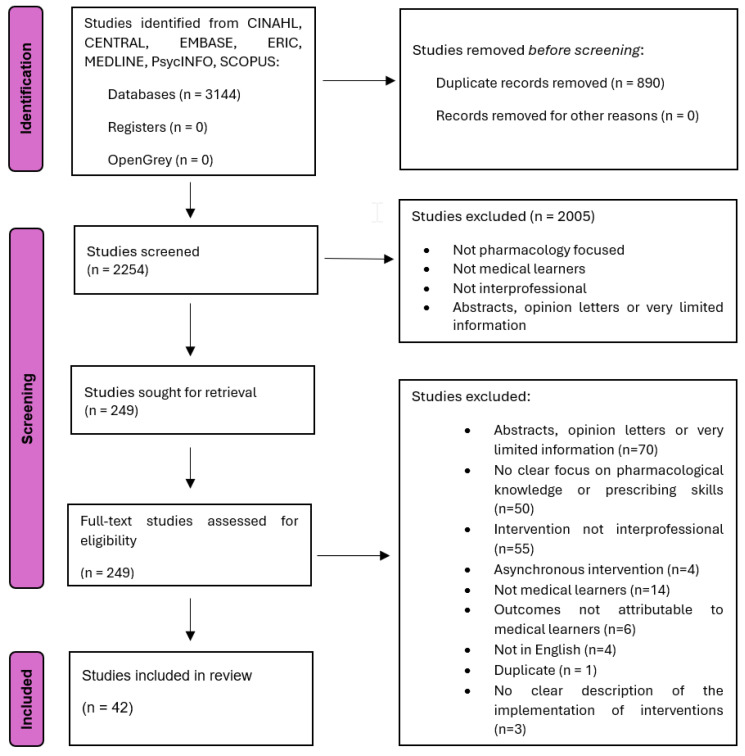
PRISMA Flow Chart.

**Table 1 pharmacy-13-00116-t001:** Kirkpatrick expanded learner outcomes for IPE.

Level 1: Learner’s Reaction	Learners’ views on the learning experience and its interprofessional nature
Level 2a: Modification of attitudes/perceptions	Changes in reciprocal attitudes or perceptions between participant groups; changes in attitudes or perceptions regarding the value and/or use of team approaches to caring for a specific client group
Level 2b: Acquisition of knowledge/skills	Including knowledge and skills linked to interprofessional collaboration
Level 3: Behavioural change	Individuals’ transfer of interprofessional learning to their practice setting and their changed professional practice
Level 4a: Change in organisational practice	Wider changes in the organisation and delivery of care
Level 4b: Benefits to patients, families, and communities	Improvements in health or well-being of patients, families, and communities

Note: Adapted from a conceptual framework for measuring the impact of IPE [24].

## Data Availability

No new data were created or analyzed in this study. Data sharing is not applicable to this article.

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
