# Peer review of "Interprofessional Educational Interventions to Improve Pharmacological Knowledge and Prescribing Competency in Medical Students and Trainees: A Scoping Review"

_pharmacy, 2025, doi:10.3390/pharmacy13050116_

Round 1
Reviewer 1 Report
Comments and Suggestions for Authors
Excellent scoping review with strong methodology which included detailed definitions of the search strategy, and inclusion and exclusion criteria.
Results section was comprehensive and flowed logically in an easily readable and understandable manner.
Discussion section included the author's interpretation of results, which strongly resonated with me throughout this section of the manuscript; specifically: lines 373-380, discussing that the inclusion of doctors-in-training (DIT) in this scoping review provided useful results, as DITs are responsible for prescribing medications, thus making it more feasible to assess higher-ordered Kirkpatrick metrics which include patient outcomes in real-world settings; lines 416-419, which discusses the potential implication of study findings that medical students were less satisfied with IPE intervention (compared to other healthcare students in the samples), perhaps due to the "doctors at the top" of the healthcare hierarchy; Lines 434-435 described the possible limitation that the search strategy di not locate all possible studies; and lines 451-454, recognizing that educational interventions are often not sustainable without refresher interventions to prevent loss of knowledge and skills. Had any of these points been lacking in the discussion, I would have suggested the authors add comments to address them in the discussion.
Conclusion was supported by data presented in the scoping review.
Reviewer 2 Report
Comments and Suggestions for Authors
The title of the manuscript is consistent with the content of the manuscript
The abstract summarizes well the review
In the Introduction section the available literature data are presented regarding the possibilities to enhance the medication safety.
Recommendation: a short summary of the local education system and the legislative framework to understand those aspects which can influence prescription errors (e.g. who can prescribe medication, does the family doctor review/check the specialist's recommendation?, the possible role of pharmacist in correction of the occurred errors, etc.).
In the Materials and methods section the study procedure is properly described, we encounter clear and understandable explanation of the criteria for selecting articles/studies.
The Results section clearly presents the results obtained.
The Table 2 contains useful summary data.
In the discussion the obtained results are supported by literature data.
The presented results can be compared with other similar studies, review (it us mentioned in the manuscript that there are other similar studies).
The References are appropriate.
Reviewer 3 Report
Comments and Suggestions for Authors
Dear Authors,
congratulations on your valuable work. Please, find here below some suggestions in order to further improve the quality of your paper.
- Major comment: Why did you decide to narrow the search since 2020? An explanation about this choice should be stated.
- The introduction needs to be updated. There are several papers published in the last 5 years that could be cited as references (e.g. https://pubmed.ncbi.nlm.nih.gov/37495272/ https://pubmed.ncbi.nlm.nih.gov/35945560/ https://pubmed.ncbi.nlm.nih.gov/39799314/)
- Please, reorganize the PICO framework presentation in the methods, as it appears now, there are some suspended phrase
Reviewer 4 Report
Comments and Suggestions for Authors
Thank you kindly for the opportunity to review your manuscript reviewing articles concerning inter professional education and students studying to be physicians.
I have several comments for review. This is your work, so use those comments that you find useful and discard the rest.
I appreciate that you took the time to provide line numbers for your manuscript, it does make it easier for us to communicate with each other
Line 21 - the numbers here do not match the numbers on the PRISMA nor in the text starting on line 181. I believe they should be the same.
Line 35 - prescribing errors are THE MOST FREQUENT CAUSE, see line 13 that says that prescribing errors are A LEADING CAUSE. Only 1 of those can be correct. As written I believe that line 13 is correct. The most common causes of medication adverse events are predictable side effects. If you want to change this to reflect the most common PREVENTABLE errors that would change the metric.
Line 40 - 91% of what? 91% of every prescribing error is made by a 1st year physician? it would seem prudent then to prohibit 1st year physicians from prescribing. 91% of every prescribing error is made by a physician? Why then aren't nurse practitioners, osteopaths, and PAs doing all of the prescribing?
Line 42 - this manuscript grossly overuses the word "however". I recommend that you review that.
Line 57 - it is interesting that pharmacy students' competence and readiness are evaluated, but physician students are not evaluated. This hubris will continue to be reflected throughout this manuscript.
Line 77 - these are designed for learning, as learning and of learning doesn't make sense to me. EPAs are designed to measure competence to practice.
Line 81 - Peer-to-Peer implies equal training. Residents to credentialed professionals is not peer-to-peer. Was this perhaps physician residents and pharmacist residents?
Line 84 - be clear in this statement. Was it only future physicians who preferred to receive feedback only from their own profession or is this a consistent finding amongst all professions?
Line 136 - black boxes are computerized data storage devices accessed after an airplane disaster. Boxed warnings exist on drug where there is a need to highlight a required risk-benefit analysis. The use of the term in this manuscript is confusing.
Line 150 - pharmacists are not allied health professionals. They are health professionals. If there is any hope that this manuscript will be helpful to all health professions, comments such as this must be corrected.
PRISMA Chart & line 181. The numbers listed in the PRISMA chart don't appear to match the numbers cited on line 181 and 182 and neither agree with the abstract.
Line 190-191 There is a terminology problem. What is the difference between a medical student and a doctor-in-training. This would not be internationally standard language
Line 248 - which students? Many professions allow students to prescribe with varying degrees of supervision. Is this comment made to highlight why the training was simulated rather than patient care?
Line 358 - there are many "medical learners" who are not doctors-in-training. The parenthetical comment is confusing and likely offensive to all medical learners who are not students hoping to be physicians.
Line 416 - Who assumes the hierarchy? This is sufficiently offensive and divisive that it should be cited to someone and listed in the references.
Round 2
Reviewer 4 Report
Comments and Suggestions for Authors
I had the pleasure of reviewing the first draft of this manuscript and I am impressed with the edits and additions you have chosen to make for this submission.
I am appreciative of your highlights to clearly demonstrate those areas that were improved.
I have no comments for your team, other than my congratulations on this presentation and best wishes for disseminating your work.